# Attitudes and Opinions of Young Gynecologists on Pregnancy Termination: Results of a Cross-Sectional Survey in Poland

**DOI:** 10.3390/ijerph17113895

**Published:** 2020-05-31

**Authors:** Kornelia Zaręba, Valentina Lucia La Rosa, Ewelina Kołb-Sielecka, Michał Ciebiera, Rosalia Ragusa, Jacek Gierus, Elena Commodari, Grzegorz Jakiel

**Affiliations:** 1I Department of Obstetrics and Gynecology, Centre of Postgraduate Medical Education, 01-004 Warsaw, Poland; ewelina.sielecka@gmail.com (E.K.-S.); grzegorz.jakiel1@o2.pl (G.J.); 2Department of Educational Sciences, University of Catania, 95124 Catania, Italy; valarosa@unict.it (V.L.L.R.); ecommoda@unict.it (E.C.); 3II Department of Obstetrics and Gynecology, Centre of Postgraduate Medical Education, 01-809 Warsaw, Poland; michal.ciebiera@gmail.com; 4Health Technology Assessment Committee, Health Directorate, University Hospital “G. Rodolico,” 95123 Catania, Italy; ragusar@unict.it; 5Faculty of Psychology, University of Economics and Human Sciences, 01-043 Warsaw, Poland; jgierus@gmail.com

**Keywords:** abortion, attitudes, opinions, physicians, pregnancy termination, TOP

## Abstract

Background: This paper aims to explore the attitudes and opinions of a group of Polish young gynecologists toward pregnancy termination. Method: An anonymous questionnaire was completed by physicians who participated in obligatory trainee courses held in 2019 in Warsaw. Results: The study group included 71 physicians with an average age of 29 years (SD 3.05). A considerable number of the physicians accepted terminations for medical reasons up to the end of pregnancy, especially in cases of lethal defects (46%) and a serious disease in the mother (34%). Only 6% of the group of gynecologists not performing terminations claimed that the procedure was contrary to their conscience, and 62% of them stated that such procedures were not performed at their hospital. Terminations would be performed by 90% of the respondents in cases of lethal defects of the fetus and by 80% if severe irreversible fetal defects were diagnosed. Conclusions: The main problem associated with the inaccessibility of pregnancy termination in Poland is not linked to individuals, meaning medical personnel and the possibility of invoking the conscience clause, but probably to the lack of approval for terminations granted by hospital supervisors. Adequate knowledge on pregnancy termination procedures, fetal defects, and diseases in the mother translated into the changes of opinions on pregnancy terminations.

## 1. Introduction

All pregnancies are associated with significant existential and emotional changes in a woman, which may reinforce pre-existing internal conflicts [1,2]. In some cases, pregnancy may be unwanted or there may be particular conditions such as fetal abnormalities or pregnancy complications that make abortion necessary [3,4]. Abortion is a very complex phenomenon, both for women and for healthcare professionals [5]. A physician’s decision to perform a pregnancy termination is affected by several factors, including the law, the healthcare organization, accessibility of health services, and the social situation of the country [6]. The accessibility of the procedure and the social situation in the country are crucial factors in considering the procedure of pregnancy termination. Legal abortion has been widely accessible in Poland since 1956. A 1993 act limited the allowable reasons for pregnancy termination to strict medical indications and sexual offences. According to the 7 January 1993, Act on Family Planning, Protection of the Human Fetus, and Conditions for Pregnancy Termination, legal abortion may be performed in the following cases: the pregnancy poses a threat to maternal health or life (up to 22 weeks of gestation); prenatal screening or other medical evidence indicates a high probability of severe and irreversible fetal anomaly or an incurable life-threatening disease (up to 22 weeks of gestation); there is reason to believe that the pregnancy is the result of an unlawful act (up to 12 weeks following conception). Such a circumstance must be confirmed by a prosecutor. According to official data, 1057 legal terminations were performed in Poland in 2017, including 22 procedures performed due to a threat to the mother’s life and health, 1035 due to severe and irreversible fetal defects, and none due to criminal offenses [7]. The number of terminations significantly varies among the provinces. Polish law is rather precise in determining the conditions of pregnancy termination, and the procedure is technically easy to perform, even by a young resident. 

Pregnancy termination is still the focus of a heated discussion both in the social aspect and among medical personnel in Poland. Numerous healthcare workers invoke the conscience clause to refuse to participate in pregnancy terminations [8,9]. A recent study showed that few prenatal care providers were willing to perform pregnancy terminations themselves [10]. Therefore, we decided to investigate the underlying reasons. The first group we analyzed consisted of young physicians during their specialization period in obstetrics and gynecology. They were a group of persons whose education took place under a new system in which indications for pregnancy termination were determined by a restrictive abortion law. They did not remember the time when abortion was widely available in our country, and they had not participated in abortion procedures on request without medical indications. The conscience clause prerogative is widely exercised by Polish healthcare workers. Gynecologists frequently invoke the conscience clause not only when refusing to terminate a pregnancy but also when prescribing contraceptives. As a consequence, the right to terminate a pregnancy due to medical indications is not observed in many places in Poland, which results in multiple pregnancy terminations being performed abroad. According to international professional literature, the Polish approach to the conscience clause is described as “conscience absolutism” [11]. The present study was designed to obtain information about whether the lack of common access to pregnancy termination results from the personal beliefs of physicians, system conditions, or the fear of stigmatization by society.

## 2. Materials and Methods

An anonymous questionnaire was completed by the physicians enrolled in obligatory courses included in a five-year residency program at the First Department of Obstetrics and Gynecology, Center of Postgraduate Medical Education in Warsaw. Physicians during the specialization period in obstetrics and gynecology who gave their consent to participate in the study were included. A physician conducting the study requested that they complete the questionnaire. The questionnaires were collected at the end of the course and 100% were returned. The study was approved by the Bioethics Committee (approval number 71/PB/2018). 

### 2.1. Questionnaire

The physicians were recruited prospectively. The questionnaire included 32 questions grouped into four sections: general information, religion, worldview, and final remarks. Some answers were presented on a 5-point Likert scale (a–“I strongly agree” through e–“I strongly disagree”) (Appendix A). Completing one questionnaire required about 10 minutes. Any participant questions were answered by the researcher. A total of 71 questionnaires were obtained. Subsequently, statistical analysis was performed.

### 2.2. Statistical Analysis

A nominal scale was used to measure the majority of variables due to the questionnaire-based character of the study. Preliminary analyses included descriptive statistics, such as frequencies and means. Statistical software was used to generate cross tables and multivariate tables. Some questions were answered with the use of the Likert scale. The responses were presented as medians and modes with standard deviation calculation. The criterion of statistical inference was set at the level of significance of *p* < 0.05. There was no assumption of intergroup differences, so no test of differences was used. Chi-squared normality tests were conducted (Shapiro–Wilk, Kolmogorov–Smirnov). All distributions were non-normal. Analyses were performed with the StatSoft Statistica 13.1 statistical package.

## 3. Results

The study group included 71 physicians aged 26 to 36 (Table 1). The average age in the group was 29 ± 3.05. The percentage of women (69%) was over twice that of men (31%). The majority of the respondents (76%) lived in a province capital and all provinces of the country were represented by participants in the study group. Of the respondents, 47% were employed at teaching hospitals. Physicians working in this profession for 1–2 years constituted the largest group (48%). The study group was rather uniform regarding marital status; 53.5% of the respondents were married and 37% had children. Approximately 79% of the respondents declared themselves as Catholics, of whom only 28% claimed to be practicing their religion, while weekly mass attendance was reported by 11%.

Fetal ultrasound was performed by approximately 96% and prenatal testing by 18% of the respondents. Despite performing prenatal testing, not all physicians specified in which facility pregnancy termination can be performed. Such information was passed along to the patients by 72% of the respondents. All respondents indicated acceptance of in vitro fertilization (IVF) procedures, with 74% opting for the necessity of state financing for the procedure. Based on the conscience clause, 14% of the physicians refused to prescribe emergency contraception.

### 3.1. Personal Beliefs Regarding Indications for Pregnancy Termination

A large number of the physicians accepted terminations for medical reasons up to the end of pregnancy, especially in cases of lethal defects (46%) and a serious disease in the mother (34%) (Table 2). Over 49% of the respondents did not approve of termination if a patient wished to discontinue the pregnancy without medical indications. Down syndrome was distinguished as a separate category due to numerous controversies regarding the indications for pregnancy termination. In this case, 23% of physicians objected to pregnancy termination. Surprisingly, 13% of the respondents (7 women and 2 men) disapproved of pregnancy termination in the case of rape.

No significant differences in relation to the sex of the respondents were observed regarding beliefs concerning pregnancy terminations, except in the most serious maternal diseases (χ2 = 0.740; *p* = 0.389). In such cases, five women disapproved of termination, while all men accepted it.

Physicians working for teaching hospitals were more likely to accept pregnancy terminations than those employed by nonteaching hospitals. The procedure of pregnancy termination was approved by 84% of teaching hospital physicians versus 77.5% of nonteaching hospital physicians (*p* < 0.05). The respective values in the case of lethal defects of the fetus were 93.5% versus 87.5% (*p* = 0.959), 77% vs. 77.5% (*p* = 0.821968) for serious maternal disease, and 74% versus 60% (*p* = 0.914) for a pregnancy resulting from rape.

Respondents who had no children opted more frequently for pregnancy termination (Table 3). Childless physicians would be willing to perform pregnancy termination more often than physicians who had children and this difference is statistically significant (χ2 = 10.259; *p* = 0.016). One striking feature of the results was the altruistic attitude of physicians in the case of a severe fetal defect in one’s own child. In the case of the physician’s own child, the pregnancy would be continued by eight gynecologists despite severe fetal defects diagnosed and by 22 gynecologists if a serious disease was confirmed in the mother. However, they would perform pregnancy termination upon request of the patient in the case of a serious disease in the mother and fetus.

### 3.2. Personal Experience in Pregnancy Termination Procedures

Pregnancy terminations were performed personally by 32% of the respondents. Only 6% of the group of gynecologists not performing terminations claimed that the procedure was contrary to their conscience, while 62% stated that such procedures were not performed at their hospital.

Regarding specific medical indications, terminations would be performed by 90% of the respondents in the case of lethal defects of the fetus and by 80% of respondents if severe irreversible fetal defects were confirmed. However, only 14% of the respondents would perform an abortion of a healthy fetus upon maternal request. It is also noteworthy that despite refusing to personally perform an abortion without indications, 41% of the respondents accepted the right of women to undergo abortion on request (Figure 1 and Figure 2). Moreover, male physicians would be more likely to perform pregnancy terminations personally (Figure 2).

### 3.3. Other Results

We also enquired about possible alternative solutions regarding pregnancy continuation in cases of maternal or fetal disease. Approximately 33% of physicians claimed that Caesarean section was an inappropriate procedure to be performed in cases of lethal fetal defects. Most gynecologists viewed Polish society as intolerant in terms of pregnancy termination (80.23%). However, none of the gynecologists claimed to refrain from performing pregnancy terminations for fear of the negative opinion in society.

## 4. Discussion

The accessibility of pregnancy termination in Poland is most similar to Australia and Italy [6,12]. Up to 70% of physicians invoke the conscience clause and refuse to perform the procedure in Italy, which is similar to Poland [12]. A questionnaire study conducted in a group of teaching hospital physicians from 16 countries showed that their views on pregnancy termination of fetuses with moderate defects were more restrictive than the law of a given country in about 33% to 50% of physicians. The majority (65%) did not opt for liberalization of the law [13]. Late terminations are not available for women in Poland. According to liberal groups, terminations before and after 22 weeks of gestation are equally ethical [14]. In the present study, the respondents approved the liberalization of abortion law and the possibility to terminate a pregnancy up to the end of the gestation in cases of lethal fetal defects (46%) and of serious maternal disease (34%). The largest group of respondents in our study (46%) opted for termination up to the end of pregnancy in the case of a lethal defect. It is highly probable that such a belief was due to their knowledge about the character of lethal defects. This is consistent with the fact that fetal ultrasound was performed by approximately 96% and prenatal testing by 18% of the respondents. The awareness of possible complications that might occur during the delivery of a gestation carried to term is obviously also significant. Approximately 33% of the physicians claimed that Caesarean section was an inappropriate procedure to be performed if lethal fetal defects were diagnosed. In Poland, the procedure of the embryotomy of the fetus carried to term in the case of lethal defects belongs to the history of medicine. The resultant question refers to an alternative solution if the decision to terminate the pregnancy was not made.

In addition to malformations, the contraction of infectious diseases that can be transmitted to the fetus [15] and the need to carry out treatments with drugs that can harm the newborn, must be taken into consideration for the termination of pregnancy. In Poland, the screening routine of hepatitis C virus (HCV)was introduced in pregnant women in 2010, to prevent mother-to-child transmission [16]. It was associated with an increase in the percentage of HCV-positive women, but there are no cases of voluntary abortion in the case of HCV infection. In the case of cancer, the mother often renounces chemotherapy, but does not ask for an abortion [17].

Professional literature also tackles the issue of a discrepancy between a physician’s certainty concerning the necessity to perform a termination and their willingness to perform the procedure personally [14]. It is encouraging that, despite the lack of consent to perform abortion without indications, 41% of young gynecologists accepted the right of women to undergo abortion on request. According to Polish law, physicians who invoke the conscience clause are obliged to inform the patient where the procedure may be conducted, but this is not always done, so the law is a dead letter in many aspects. Sepper stated that such an approach delayed the procedure of pregnancy termination and additionally traumatized the patient who was seeking help in another facility [18]. In many countries, the procedure is performed by qualified midwives; in numerous other countries, early pregnancy is terminated by family doctors [19,20,21,22]. A comparative multiple-case study conducted in four European countries (Italy, Portugal, England, and Norway) showed that the right to invoke the conscience clause is only granted to those directly involved in the procedure of pregnancy termination. It does not apply to persons who perform prenatal tests, as they are obliged to refer the patient to a center in which it is possible to undergo the procedure [9]. The conscience clause may be also used by an anesthesiologist performing anesthesia for termination of pregnancy [23]. One Australian expert described the refusal to terminate a pregnancy, thus imposing one’s views on the patient, as “a slap in the face (to those women)” [24].

Another factor that influences the decision to perform pregnancy termination is associated with the physicians’ fear of facing repressive behaviors from the medical environment and society. A comparative multiple-case study conducted by Italian physicians in four European countries showed that persons performing pregnancy terminations are discriminated against, their burden of work is increased, and they are limited in terms of professional advancement [9]. Similarly, Australian physicians expressed their concern about social repression connected with being categorized as an “abortion doctor” [19,24]. An internet questionnaire study conducted among physicians performing pregnancy terminations in the USA demonstrated that 48.2% of physicians performing abortions had faced at least one malpractice claim, disciplinary action, or a criminal charge, and had at least 30 years of job seniority [25]. It is possible that the older age of American physicians was associated with a lesser fear of social ostracism, which was reported by Italian physicians. The present study showed that the respondents, despite their young age, were not concerned about repression from society or the environment.

Professional knowledge on the issue is of great importance regarding the assessment of indications for pregnancy termination. During a conference held in 1999 in Sydney, a study was conducted to assess the attitude of physicians, members of the Society of Human Genetics, to pregnancy termination at its various stages. Over 80% of geneticists (82%) and obstetricians performing ultrasound examinations (88%) would decide to terminate a pregnancy at 24 gestational weeks due to acrania and over 75% due to spina bifida. In the case of Down syndrome, termination at 24 gestational weeks would be offered by 72% of geneticists and 33% of obstetricians [26]. According to a study conducted in the Republic of South Africa, 51% of medical students stated that they had liberalized their views on pregnancy termination in the course of their medical studies [8]. Individuals who had already initiated their sexual life (a group that is directly interested in complications associated with sexual life) had more extensive knowledge regarding abortion. It is another piece of evidence confirming that personal experience influences both seeking to acquire new knowledge and a possible change of opinion afterwards. Furthermore, this is an important argument in favor of introducing essential knowledge about the law and conditions of pregnancy termination in Poland during sex education classes. A study conducted in medical students in Ireland, a country with more restrictive abortion laws than Poland’s at the time the study was conducted, revealed that 72.8% of the respondents claimed to have a prochoice attitude [19]. Only 7.1% of the respondents disapproved of abortion, while 55% approved of abortion on request. They also stated that “laws against abortion don’t stop abortion, they simply make it safe” (68.9%) [27]. Interestingly, the religious aspect of termination was mentioned only by 21.1% of the respondents in this predominantly Catholic country. Only 6% of physicians invoked religious considerations in the present study. Apparently, opinions concerning pregnancy termination undergo fundamental transformation over time. A comparative study conducted in 2009 among students at the University of Oslo and Queen’s University Belfast showed that only 14.3% of Irish students were in favor of abortion [28]. A study conducted in 23 medical students in India revealed that the lack of knowledge regarding the procedure and the current law of the country contributed to their disapproval of pregnancy terminations [29]. Moreover, appropriate knowledge will support primary care providers and nurses in delivering medical abortion services and increasing access for women [30].

According to professional literature, the main problem in the USA and the Republic of South Africa is associated with systemic issues [31,32]. The presented questionnaire showed real reasons for refusing to perform terminations. The study demonstrated that pregnancy terminations were not performed in numerous provinces because of the beliefs of the heads of the health facilities. Only three physicians stated that pregnancy termination was out of line with their convictions, but 31 (62%) reported that no terminations were performed at their facility. The present study showed that terminations would be performed by 90% of the respondents in cases of lethal defects of the fetus and by 80% of respondents in cases of severe irreversible fetal defects. Therefore, it may be assumed that the change in the attitude of the heads of healthcare centers would diametrically shift the situation of pregnancy terminations in Poland.

### Strengths and Limitations

The present study marks the first time the younger generation addressed the topic of fetal defects and pregnancy termination procedures. Participants were physicians who are most aware and who possess practical and theoretical knowledge of the aforementioned aspects. The study encompassed physicians of all provinces in Poland and revealed a systemic problem whose existence of which we were unaware. All the physicians were eager to complete the questionnaire (the return rate was 100%) and claimed that the issue was important and required many changes. They were very pleased that they were able to express their opinion on the subject.

Regarding the limitations of our study, it is based on a questionnaire that probably did not tackle numerous significant issues but suggests a foundation for further research. New groups to be included in the project will consist of older gynecologists and physicians whose profession is to perform fetal diagnostic work-up. Questions about beliefs concerning the termination of one’s own pregnancy may not be fully reliable because numerous studies show that one’s perspective on pregnancy termination changes when a problem becomes personal. Because the present study was cross-sectional, it is not possible to assess whether the respondents with greater job seniority changed their viewpoint regarding pregnancy termination. Moreover, respondents are predominantly nonpracticing Roman Catholics and the group is not representative for contemporary Polish values, but instead representative of contemporary values for college educated or medical school educated Polish people.

## 5. Conclusions

The main problem associated with the inaccessibility of pregnancy terminations in Poland is not linked to an individual (medical personnel) and the possibility of invoking the conscience clause, but to an institutional problem. The underlying reason will be another topic for our research. It may be assumed that it is due to the fear of social stigmatization or imposing the views of the administrators on the whole team. Systemic changes should constitute the main assumption of a new state policy concerning pregnancy termination. Institutional conscience does not exist, and the study showed that physicians approved of pregnancy termination. Moreover, they are willing to perform such a procedure personally in cases indicated by the law.

The study showed that adequate knowledge of pregnancy termination procedures, fetal defects, and diseases in the mother translated into the liberalization of opinions on pregnancy terminations. Knowledge about possible complications associated with the delivery of a pregnancy carried to term, prognosis concerning fetal defects, and the procedure itself contribute to the changes of one’s viewpoint on the appropriate indications for pregnancy termination [5]. It is also important to offer adequate contraceptive counseling to the woman during her request for abortion, in order to avoid other unplanned pregnancies [33].

It is of interest that the cohort of younger physicians is largely devoid of fear of environmental pressure, and their reasoning is independent from the systemic resistance associated with the controversial topic of pregnancy termination. Hopefully, this will contribute to changes in future indications for pregnancy termination and accessibility of the procedure.

## Figures and Tables

**Figure 1 ijerph-17-03895-f001:**
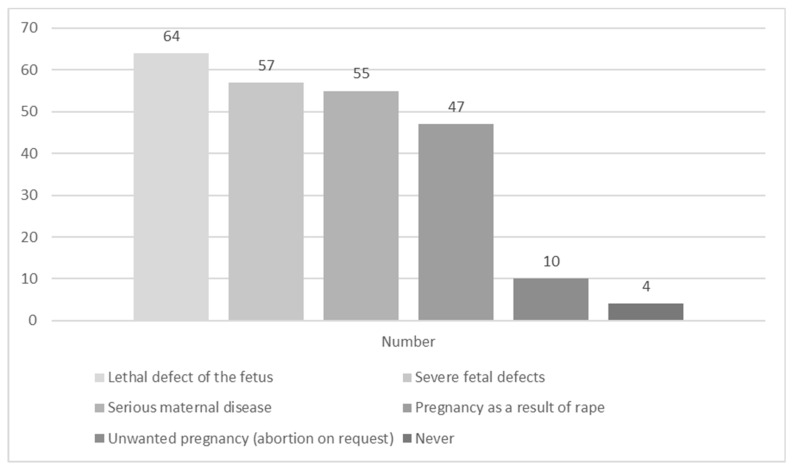
Situations in which physicians would be inclined to perform pregnancy terminations personally.

**Figure 2 ijerph-17-03895-f002:**
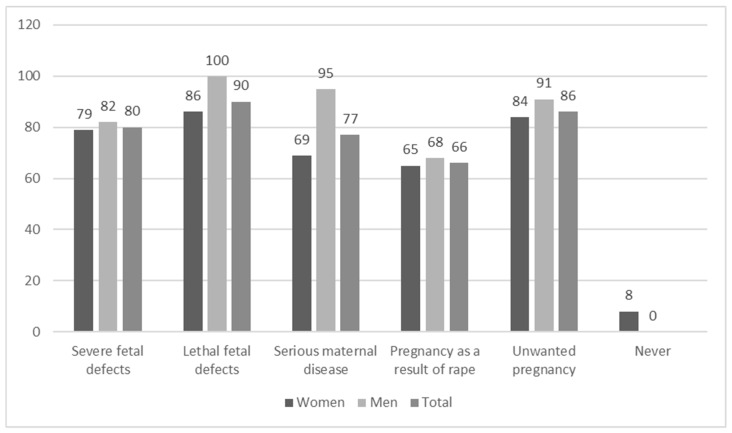
Declaration to perform pregnancy termination depending on the respondent’s sex (in %).

**Table 1 ijerph-17-03895-t001:** General description of the group**.**

Variable		n = 90
**Age**		29.16 ± 3.05
**Sex**	Women	49 (69.01%)
	Men	22 (30.99%)
**Size of the town/city**	Province capital	54 (76.06%)
	District capital	15 (21.13%)
	Other towns	2 (2.81%)
**Province**	North Poland	15 (10.65%)
	South Poland	11 (7.81%)
	East Poland	13 (9.23%)
	West Poland	4 (2.84%)
	Central Poland	27 (19.17%)
**Type of hospital**	Teaching hospital	31 (43.66%)
	Provincial hospital	22 (30.98%)
	District hospital	9 (12.68%)
	Another hospital	9 (12.68%)
**Marital status**	Unmarried	33 (46.48%)
	Married	38 (53.52%)
**Children**	No	44 (61.98%)
	Yes	26 (36.63%)
**Job seniority in gynecology**	<1 year	9 (12.68%)
	1–2 years	34 (47.89%)
	3–5 years	20 (28.17%)
	6–10 years	5 (7.04%)
	>10 years	2 (2.82%)
**Religion**	Catholicism	56 (78.87%)
	Islam	2 (2.82%)
	Agnostic	10 (14.08%)
	Others	3 (3.22%)

**Table 2 ijerph-17-03895-t002:** Approval of pregnancy terminations in individual clinical situations.

	Until the End of Pregnancy	Until 22 Gestational Weeks	Until 12 Gestational Weeks	No Approval	No Information
**Abortion without medical indications**	1 (1.4%)	5 (7.04%)	29 (40.84%)	35 (49.3%)	1 (1.4%)
**Severe fetal defects**	20 (28.16%)	41 (57.74%)	4 (5.63%)	5 (7.04%)	1 (1.4%)
**Lethal fetal defects**	46 (46.47%)	32 (45.07%)	3 (4.22%)	3 (4.22%)	0 (0%)
**Rape**	12 (16.90%)	24 (33.80%)	26 (36.61%)	9 (12.67%)	0 (0%)
**Serious life-threatening maternal disease**	24 (33.80%)	11 (15.49%)	30 (42.25%)	5 (7.04%)	1 (1.40%)
**Down syndrome without anatomical anomalies**	8 (11.26%)	38 (53.52%)	8 (11.26%)	16 (22.5%)	1 (1.40%)

**Table 3 ijerph-17-03895-t003:** Attitude towards pregnancy termination depending on having children.

	Severe Fetal Defects	Lethal Fetal Defects	Serious Maternal Disease	Pregnancy as a Result of Rape	Unwanted Pregnancy	Never
Childless physicians	60 (84%)	68 (95%)	57 (79%)	50 (70%)	12 (16%)	4 (4.5%)
Physicians who had children	55 (77%)	61 (85%)	45 (77%)	44 (61.5%)	8 (11.5%)	7 (7.5%)
Total (absolute frequency)	58 (81%)	65 (91%)	56 (78%)	48 (66.5%)	11 (14.5%)	5 (5.5%)

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
