# Peer review of "Attitudes and Opinions of Young Gynecologists on Pregnancy Termination: Results of a Cross-Sectional Survey in Poland"

_ijerph, 2020, doi:10.3390/ijerph17113895_

Round 1
Reviewer 1 Report
This original cross sectional survey research found young Polish Ob Gyns in training were more willing to perform abortions, particularly for lethal fetal abnormalities, than they were permitted to do in their hospitals. Importantly, this documents also that the respondents are predominantly nonpracticing Roman Catholics. The article does not evaluate whether this is representative of contemporary Polish values, or contemporary values of college educated or medical school educated Polish people.
Reviewer 2 Report
Well organized study and important topic. Overall, I think that this article should be published. Table 1 needs reformatting--it is awkward to read. The 100% response rate needs explanation--it is so good that it is hard to believe. That will give readers pause unless there is an explanation. Need to explain or rephrase a new generation of physicians. Cohort would be a better word. Just because the respondents are from all over the country doesn't mean that you have a truly representative sample. Needs explanation. Are there differences between rural and urban physicians? Are these physicians different from the older cohort?Author Response
Attached
